# RoboReport: Summarizing a virtual robot's past actions in natural language

**Chad DeChant**
Computer Science Department
Columbia University
chad.dechant@columbia.edu

**Iretiayo Akinola**
NVIDIA Corporation
iakinola@nvidia.com

**Daniel Bauer**
Computer Science Department
Columbia University
bauer@cs.columbia.edu

**Abstract:** We demonstrate the task of giving natural language summaries of the actions of a robotic agent's actions in a virtual environment. Existing datasets that match robot actions with natural language descriptions designed for instruction following tasks can be repurposed to serve as a training ground for robot action summarization. We propose and test several methods of learning to generate such summaries, starting from either egocentric video frames of the robot taking actions or text representations of the actions and find a two stage summarization process which uses structured language as an intermediate step improves accuracy. Quantitative and qualitative evaluations of the results are provided to serve as a baseline for future work.

**Keywords:** RoboNLP, Interpretability, Summarization

## 1 Introduction

As robots become more capable and are entrusted with more tasks, it will be increasingly important to reliably keep track of what they do. However, robots will routinely perform roles that make direct supervision of them difficult or impossible. A robot may, for example, be used to move many loads of construction material from place to place or perform household chores. In both cases, real time human oversight would be impractical. It will therefore be necessary to develop methods to monitor and record the actions of such agents and provide that information at a later time to a human. One way to do that is to develop the capability for robots to report on and summarize their actions in natural language. Summaries, rather than complete records, will be particularly useful as action sequences become longer. They will also be challenging to produce because it will be necessary to identify the most important actions and, very often, to describe those actions using higher level abstract terms.

The task of robot action summarization is a new one and there is no existing dataset explicitly designed for it. We suggest that popular datasets of natural language instruction following for robots [1, 2, 3] can be repurposed for robot action summarization. For example, the popular ALFRED dataset [1] is used to train robotic agents in the virtual AI-2 Thor [4] environment to perform complicated, multi-step tasks given natural language instructions. We propose inverting that order, instead using it to train an agent to summarize the actions it takes while performing the tasks.

We assess the accuracy of producing summaries from different input modalities such as natural language, formal plans, or images. First, if a robotic agent may be able to produce a record of its actions in the form of short keywords, we find that these can very accurately be transformed into a summary. Second, if such text is unavailable, we show that we are able to generate summaries using frames from egocentric videos captured by the agent, though these summaries are not as accurate as those derived from text. Third, we find that these summaries can be improved if we use a two

Workshop on Language and Robot Learning
6th Conference on Robot Learning (CoRL 2022), Auckland, New Zealand.

| | Generated text | Ground truth text |
|---|---|---|
| **Summary** | put a *glass* in a sink. 
 put a clean rag in the sink. 
 put two *tomatoes* on the counter | Put a *bottle on a plate* in the sink. 
 Place a clean rag on a counter. 
 To **cut** *apples* on the counter with a knife |
| **PDDL** | gotolocation shelf pickupobject bowl gotolocation countertop putobject bowl countertop pickupobject egg putobject egg *plate* | gotolocation shelf pickupobject bowl gotolocation countertop putobject bowl countertop pickupobject egg putobject egg *bowl* |
| **Actions** | lookdown moveahead moveahead moveahead moveahead moveahead ~~pickupobject~~ pickupobject bread **rotateleft** moveahead moveahead moveahead moveahead moveahead moveahead lookdown openobject fridge putobject bread fridge closeobject fridge openobject fridge pickupobject bread closeobject fridge lookup **rotateleft** moveahead moveahead moveahead moveahead moveahead moveahead ~~moveahead moveahead~~ putobject bread countertop | lookdown moveahead moveahead moveahead moveahead moveahead ~~rotateleft lookup~~ pickupobject bread ~~lookdown~~ rotateright moveahead moveahead moveahead moveahead moveahead ~~rotateright~~ lookdown openobject fridge putobject bread fridge closeobject fridge openobject fridge pickupobject bread closeobject fridge lookup rotateright moveahead moveahead moveahead moveahead moveahead moveahead ~~rotateleft~~ moveahead putobject bread countertop |
| **Instructions** | walk to the coffee pot. grab the coffee cup that is under on coffee pot. carry the cup to the microwave . put the coffee cup in the microwave. take the coffee cup out of the microwave. carry the cup back to the coffee pot. | ~~Turn around~~ and walk to the coffee maker on the counter. Pick up the coffee mug from the coffee maker. ~~Turn left~~ and walk to the microwave. Heat the mug in the microwave ~~Turn around~~ and walk to the coffee maker. Put the mug in the coffee maker. |
| **Key** | **Action errors**    *Object errors*    Place errors | ~~Extra errors~~ (shown in generated text)    ~~Omission errors~~ (shown in ground truth text) |

Figure 1: Examples of system outputs of natural language summaries, PDDL, low level actions, and natural language instructions generated from video frames. Examples are selected to show representative errors, which are annotated according to the key at the bottom of the figure.

step pipeline which takes in the video frames, first produces a text description of actions in an intermediate structured language, and then transforms that text into natural language summaries.

## 2 Method

**Problem definition** Our objective is to generate summary report in natural language $l \in \mathcal{L}$ of a long horizon robotic task, given the history of observations $o \in \mathcal{O}$, actions $a \in \mathcal{A}$ or intermediate plans $p \in \mathcal{P}$ that the robot experienced during the task. We define the robot experience/trajectory as $\tau = \{(o_0, a_0, p_0), ...\}$. We seek to learn a function $\mathcal{F}_\theta$ such that: $l = \mathcal{F}_\theta(\tau)$.

**Repurposed dataset** An episode of action in the ALFRED dataset [5] we use has five different kinds of representation: 1) *Summaries*: Human-generated natural language one sentence summaries of the whole action sequence (called "goal descriptions" in the original dataset). 2) *Instructions*: Natural language step by step instructions over several sentences written by humans. 3) PDDL: High level action plan in the Planning Domain Description Language [6] with semantically rich content. 4) *Action descriptions*: Low level action plan generated by an automatic planner corresponding to available actions in the environment. It is more detailed and longer than the PDDL but less easily readable and contains slightly less semantically rich content. For example, where the higher level PDDL might read "GotoLocation alarmclock" the lower level actions might be a sequence of "MoveAhead", "RotateRight", and "RotateLeft" actions. 5) *Video, images, and visual features*: Raw video of a task episode as well as still frames from the video, including a pre-selected subset of such frames which we use here, and 512 7 × 7 layers of features of those frames extracted from a Resnet-50 convolutional neural network [7].

**Single stage summarization models:** To generate summaries from text, we experiment with using three kinds of text as input: PDDL, lower level action plans, and natural language instructions. For all of these experiments, we fine-tune a T5 large language model transformer originally trained on multiple tasks including summarization [8] ("t5-base" from the Hugging Face library [9]). To study going from visual input to text descriptions, we pass the Resnet image features through two further convolutional layers trained from scratch and then into a bidirectional recurrent encoder-decoder network with attention, also trained from scratch, to output text, using a cross-entropy loss over the vocabulary present in the dataset.

| Input | No errors | Action errors | Object errors | Place errors | Extra errors |
|---|---|---|---|---|---|
| PDDL | 98% | 0% | 0% | 2% | 0% |
| Actions | 96% | 0% | 4% | 0% | 0% |
| Instructions | 98% | 0% | 0% | 2% | 0% |
| Video frames | 38% | 4% | 26% | 22% | 4% |
| *Generated pddl* | *54%* | *10%* | *8%* | *38%* | *4%* |

Table 1: Manual error analysis of high level summaries by method of generation. The percentage of generated examples with no errors is shown in the left column; in the right columns are percentages of errors present by type of error. Some examples may have more than one error. All examples are from the unseen validation set. Italicized last line is our two stage pipeline approach.

**Two stage summary generation via auxilliary supervision with intermediate plan :** The task of robot action summarization from a stream of images is challenging as it entails integrating high dimensional information over time into a succinct natural language summary. To address this challenge, we propose a method that uses intermediate PDDL action plans as auxilliary prediction targets to extract valuable features from images as an intermediate output using the convolutional and recurrent network described above. These generated PDDL representations are then passed through a T5 transformer which we previously fine-tuned to summarize from PDDL.

## 3 Results

Our experiments are designed to address a few questions: 1) Can high level plans (e.g. PDDL) or lists of lower level actions be used to generate natural language summaries? 2) Can we directly generate text descriptions, either structured representations or natural language, using only egocentric video frames of the robot during action? 3) Can we use the structure and relative semantic richness of PDDL descriptions as an intermediate text representation between video frames and natural language summaries? 4) How accurate will these various techniques be and what are the patterns of failure? To answer these questions, we devise automatic  manual evaluation metrics, as well as examples of our models' outputs.

First, we manually inspect a randomly selected set of fifty examples for each summary output and calculate what percentage of these are error free or have one or more errors. There are four types of error used for our analysis: *action errors*, in which the main action that took place during the relevant episode is reported incorrectly in the generated summary; *object errors*, in which the main object interacted with is wrongly named or counted; *place errors*, in which the place or places mentioned are incorrect; and *extra errors*, in which details which are unwarranted by the input are hallucinated by the model. These analyses can be found in Table 1.

Second, we automatically calculate ROUGE [10] and BLEU [11] scores for all types of output text. These automatic evaluation scores can be found in Table 2. We report ROUGE-1 (recall), ROUGLE-L F1, BLEU and BLEU-1 scores.

Third, in Figure 1 we provide examples of the generated text from video frames for each level of description (i.e. summary, PDDL, actions, or step by step instructions). We provide examples here of the kinds of errors enumerated in Table 1 as well as *ommission errors*, in which details mentioned in the ground truth text are missing in the generated text.

**Text to summary** The T5 model does a very good job of generating summaries based on either high level PDDL, low level actions, or natural language instructions as input. It makes an error on only one of our fifty manually inspected PDDL to summary examples and two in the low level actions to summary examples. The BLEU and ROUGE scores for each of these text inputs are similar and high.

**Video frame to text** Going directly from video frames to natural language summaries is less successful, as can be seen in the lower scores for both tasks in Table 2. It is also evident from the higher number of errors flagged by the manual inspection of the generated summaries in Table 1. Generating summaries from video frames alone has by far the lowest rate of success; only 38% of

| Task | Seen | | | | Unseen | | | |
|------|------|------|------|--------|------|------|------|--------|
| | R-1 | R-L | Bleu | Bleu-1 | R-1 | R-L | Bleu | Bleu-1 |
| PDDL to summary | .628 | .590 | .624 | .902 | .610 | .587 | .607 | .890 |
| Actions to summary | .610 | .589 | .604 | .881 | .630 | .599 | .647 | .900 |
| Instructions to summary | .624 | .596 | .663 | .917 | .629 | .598 | .665 | .910 |
| Images to PDDL | .923 | .923 | .854 | .942 | .761 | .763 | .594 | .824 |
| Images to actions | .858 | .713 | .652 | .856 | .822 | .769 | .590 | .812 |
| Images to instructions | .540 | .496 | .501 | .805 | .536 | .460 | .438 | .769 |
| Images to summary | .582 | .556 | .550 | .862 | .519 | .496 | .438 | .779 |
| *Img to pddl to summ* | *.596* | *.565* | *.580* | *.877* | *.518* | *.505* | *.472* | *.810* |

Table 2: ROUGE and BLEU scores for generated text. Results from virtual environments seen during training are on the left; unseen environments are on the right. The italicized last line is our two stage images to PDDL to summary pipeline generation approach.

inspected generated summaries contain no errors at all. Generating natural language instructions directly from frames is also challenging, likely exacerbated by the relative length and diversity of vocabulary in the instructions. By contrast, generating PDDL and low level actions is significantly more reliable than generating natural language from images.

From the sample outputs in Figure 1 it is evident that common errors include visual identification errors such as mistaking a *bowl* for a *plate*, confusing rotation directions, and missing or extra *moveahead* action commands. The fact that the BLEU and ROUGE scores for these is meaningfully worse in the unseen environments suggests that the errors derive mostly from the vision system and could be due to particular aspects of the virtual environment which may make some objects, particularly small ones, hard to recognize.

**Two stage video frame to summary via intermediate representation** Because the PDDL generated from video frames is of significantly higher quality than the generated natural language summaries or instructions from the same frames, we developed a pipeline that first takes in image features and produces the corresponding PDDL before producing summaries. We find that according to both automatic metrics and manual inspection of summaries, using the generated PDDL as an intermediate step produces better output than is seen when going from video frames directly to summaries or instructions. PDDL is less expressive than free form English so perhaps forcing the system to use it to represent action sequences provides fewer opportunities to make mistakes than going directly to natural language.

## 4 Related work

DeChant and Bauer [12] proposed and argued for the importance of enabling robots to learn to summarize their past actions. Some work on following instructions in simulated environments for navigation and task performance has used the generation of natural language descriptions of navigation trajectories as a training signal or tool: Nguyen et al. [13] provide feedback to an agent in the Room to Room environment by describing in natural language the paths the agent actually takes so it can learn to compare that to the path it should have taken; Fried et al. [14] learn to generate instructions to augment training data and then, at test time, to evaluate the similarity of routes it might take with the description of the route it is supposed to take. The challenges of natural language summarization have been addressed for many years in the NLP literature [15, 16]. Recurrent sequence to sequence models have been employed for so-called abstractive summarization, which outputs newly generated text rather than simply selecting text, [17, 18], as have transformer architectures [19, 8].

## 5   Conclusion

This work begins a line of research on robot action summarization. It is important that robots operating in the real world be well supervised by humans and that their actions be understandable. We suggest that establishing a basic narrative of *what* an agent does is in some ways a prerequisite to understanding *why* it does something. Once action summarization can be performed reliably, we expect it to be useful in a variety of ways, including in the training of robotic agents, automatically generating labels for action sequences, and in lifelong learning settings in which robots might receive feedback to the summaries they generate.

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
