# OpenReview forum: "RoboReport: Summarizing a virtual robot's past actions in natural language"
_robot-learning.org/CoRL/2022/Workshop/LangRob — LangRob 2022 Spotlight_

### Official Review · Reviewer_Jm9o · 2022-11-13

**Rating:** 8
**Confidence:** 5

**Review:**

This is a really cool paper that presents a very interesting finding. The authors do a great job at conveying the core hypothesis clearly and presenting a suite of approaches that a reader may wonder as possible solutions. Despite being short, I found the paper to be complete and answered many initial questions that cropped up.

Most importantly, the finding of "image to PDDL to summary" makes a lot of sense and is a great insight for future work in summarizing agent behavior. An interesting application of something like this could be in:

- generating natural language scene description for text-based models/LLMs for planning without using visual-language models. At present, people use naive visual captioning or CLIP-like models to summarize the current image frame, but something like this could be a really good fit to condition text-based planning models on the history of the robot's observations

- scene descriptions for visually impaired people. It's a far reach, but this could make a great accessibility tool.

---

### Official Review · Reviewer_uYGH · 2022-11-14

**Rating:** 7
**Confidence:** 4

**Review:**

Summary: The authors propose the task of robot action summarization in natural language. With increased robotic deployment in the real world for longer horizon tasks, such interpretable summaries may become increasingly useful. The authors propose learning summarization models from different text inputs (PDDLs, lower level cation plans, human-annotated instructions) and also investigate incorporating visual inputs.

Strengths: The proposed task setting is (to the best of my knowledge) novel and interesting, and likely to be increasingly relevant. The proposed method also presents a number of baselines investigating different accessible text-based inputs as well as different methods for incorporating images. The idea of using a generated PDDL as an intermediate step between image inputs and summary is also interesting and the initial results seem promising.

Weaknesses / Feedback:  Having identified the different error categories, the paper would be strengthened if the authors further investigate and propose methods for addressing some of the different error categories. It seems like generating a text summary from text-based inputs (PDDL, actions, instructions) is already very unlikely to be error-prone, but perhaps the more realistic setting is to focus on improving summarizations from video frames (e.g. from the robot's egocentric camera view) and to compare more directly with relevant methods like video captioning approaches.

Minor note:
- Table 1 is a little difficult to interpret at the moment -- maybe add a percentage sign so it's clearer what the numbers indicate. Convention is to also bold the best performing method, which can make Table 1 and 2 confusing.

---

### Decision · Program_Chairs · 2022-11-15

Accept (Spotlight)